Microbiology
Spectrum
# Mutual Effects of Single and Combined CFTR Modulators and Bacterial Infection in Cystic Fibrosis

Cristina Cigana,[a] Ruggero Giannella,[a] Alice Colavolpe,[a] Beatriz Alcalá-Franco,[a] Giulia Mancini,[a] Francesca Colombi,[a] Chiara Bigogno,[b] Ulla Bastrup,[b] Giovanni Bertoni,[c] Alessandra Bragonzi[a]

[a]Infections and Cystic Fibrosis Unit, Division of Immunology, Transplantation and Infectious Diseases, IRCCS San Raffaele Scientific Institute, Milan, Italy
[b]Aphad Srl, Buccinasco, Italy
[c]Department of Biosciences, Università degli Studi di Milano, Milan, Italy

Cristina Cigana and Alessandra Bragonzi share senior authorship.

**ABSTRACT**  Cystic fibrosis transmembrane conductance regulator (CFTR) modulators improve clinical outcomes with varied efficacies in patients with CF. However, the mutual effects of CFTR modulators and bacterial adaptation, together with antibiotic regimens, can influence clinical outcomes. We evaluated the effects of ivacaftor (IVA), lumacaftor (LUM), tezacaftor, elexacaftor, and a three-modulator combination of elexacaftor, tezacaftor, and ivacaftor (ETI), alone or combined with antibiotics, on sequential CF isolates. IVA and ETI showed direct antimicrobial activities against *Staphylococcus aureus* but not against *Pseudomonas aeruginosa*. Additive effects or synergies were observed between the CFTR modulators and antibiotics against both species, independently of adaptation to the CF lung. IVA and LUM were the most effective in potentiating antibiotic activity against *S. aureus*, while IVA and ETI enhanced mainly polymyxin activity against *P. aeruginosa*. Next, we evaluated the effect of *P. aeruginosa* pneumonia on the pharmacokinetics of IVA in mice. IVA and its metabolites in plasma, lung, and epithelial lining fluid were increased by *P. aeruginosa* infection. Thus, CFTR modulators can have direct antimicrobial properties and/or enhance antibiotic activity against initial and adapted *S. aureus* and *P. aeruginosa* isolates. Furthermore, bacterial infection impacts airway exposure to IVA, potentially affecting its efficacy. Our findings suggest optimizing host- and pathogen-directed therapies to improve efficacy for personalized treatment.

**IMPORTANCE**  CFTR modulators have been developed to correct and/or enhance CFTR activity in patients with specific cystic fibrosis (CF) genotypes. However, it is of great importance to identify potential off-targets of these novel therapies to understand how they affect lung physiology in CF. Since bacterial infections are one of the hallmarks of CF lung disease, the effects (if any) of CFTR modulators on bacteria could impact their efficacy. This work highlights a mutual interaction between CFTR modulators and opportunistic bacterial infections; in particular, it shows that (i) CFTR modulators have an antibacterial activity *per se* and influence antibiotic efficacy, and (ii) bacterial airway infections affect levels of CFTR modulators in the airways. These findings may help optimize host- and pathogen-directed drug regimens to improve the efficacy of personalized treatment.

**KEYWORDS**  cystic fibrosis, CFTR modulator, Pseudomonas aeruginosa, Staphylococcus aureus, antibiotic, murine model, pharmacokinetics

Address correspondence to Cristina Cigana, cigana.cristina@hsr.it, or Alessandra Bragonzi, bragonzi.alessandra@hsr.it.

The authors declare no conflict of interest.

Cystic fibrosis (CF) is caused by mutations in the cystic fibrosis transmembrane conductance regulator (CFTR) gene and affects mostly the lung, but also other organs. Respiratory failure in CF is caused by pulmonary infections and inflammation (1), with

*Pseudomonas aeruginosa* and *Staphylococcus aureus* being recognized as the most prevalent pathogens. Their chronic persistence and pathogenicity are associated with their adaptation to CF airways (2).

Traditionally, treatments have focused on lessening infection and inflammation of CF disease, although the efficacy of antibiotic therapy is not optimal. Recently, new approaches to correct CFTR cellular misprocessing (correctors) and to restore its channel function (potentiators) have emerged (3). The CFTR potentiator ivacaftor (IVA) was introduced to target gating and other residual function deficits (4–6), while the CFTR correctors lumacaftor (LUM), tezacaftor (TEZ), and elexacaftor (ELX) were approved to correct the F508del-CFTR protein and were recently extended to correct other mutations causing processing and trafficking defects. Since the most common F508del-CFTR requires both correction and potentiation for clinical efficacy, two dual-agent modulators (LUM-IVA and TEZ-IVA) (7–11) and a triple-agent drug (ELX-TEZ-IVA, named ETI) (12) have been approved. Despite the clinical benefits of these treatments, marked variability in CF patient responses has been reported (4, 5, 7, 13). This supports the view that the response to CFTR-directed therapeutics is multifactorial and that CFTR-independent factors may contribute to treatment efficacy, but the mechanisms remain unknown.

There continues to be a substantial gap in our understanding of whether and how CFTR-directed therapies impact the microbiological profile in treated patients (14, 15). IVA treatment showed an initial reduction in the prevalence of bacterial isolation, but *P. aeruginosa* density rebounded after 1 year (16), and *P. aeruginosa* strains present before treatment were found to persist even after intensive antibiotic therapy (17). Recent studies based on airway microbial metagenomic sequencing showed that ETI combination therapy reduced the load of *P. aeruginosa* and *S. aureus* in both mild and severe CF patients (18). This microbiological shift is attributable to the ETI-mediated gain of CFTR activity and the possible effect on the airway microbiota that is converted into a community of commensals. Furthermore, CFTR modulators could also have a direct effect on major CF pathogens. *In vitro* studies have shown the anti-*S. aureus* activity of IVA and its synergy with anti-*P. aeruginosa* antibiotics (19–24). LUM and TEZ have also been shown to enhance the activity of polymyxin B (PMB) against some *P. aeruginosa* isolates (22), while the impacts of ELX and ETI remain largely unexplored. However, few data have been reported on the isolates tested in those studies or their origin, particularly on the colonization stage at which they were collected and whether or not they showed adaptation traits to the CF lung. Thus, it is still unclear whether the activities of CFTR modulators differ in CF strains associated with early and advanced stages of lung colonization.

To monitor possible interactions between CFTR modulators and CF pathogens, it is critical to know the drug concentration in the lung, particularly in the epithelial lining fluid (ELF), where bacteria are present (25). Most pharmacokinetic (PK) studies have evaluated the amount of CFTR modulators and their metabolites in the plasma of different hosts (26, 27). Recently, a study evaluated IVA concentration in plasma and nasal cells from CF patients under treatment (28). However, it remains unclear the amounts of CFTR modulators that reach the airway lumen and whether bacterial pneumonia can alter their PK profiles. Indeed, only a few data on human sputum have been reported (29). Considering that most CF patients are likely colonized by bacteria when CFTR modulator therapy is initiated (14), there is a clinical need to assess the impact of infection on drug metabolism to determine the correct dosage. It is recognized that the PK of many compounds can be altered by the presence of infection (30). Recently, it has been shown that TEZ, ELX, and IVA are metabolized by cytochrome P450 3A, whose enzymatic activity is influenced by environmental factors including infection and are sensitive to inhibition and induction (31, 32). So far, the PK of CFTR modulators was assessed in normal, healthy animals, with no additional assessments conducted in the infected models (26, 27), indicating a gap between clinical manifestations of CF disease and preclinical studies. Mouse models are not helpful for efficacy studies, since cross-species comparative studies have highlighted differences between human and

**TABLE 1** MIC$_{90}$ values for IVA, LUM, TEZ, ELX, and the triple combination ETI against *S. aureus* isolates collected from CF patients at different stages of colonization (early and late)[a]

| CFTR modulator | MIC$_{90}$ ($\mu$g/mL) for *S. aureus* isolate (stage)[b] | | | | | | |
|---|---|---|---|---|---|---|---|
| | Newman (ref) | A10 (early) | A12 (late) | J6 (early) | J9 (late) | F1 (early) | F5 (late) |
| IVA | 4 | 2 | 4 | 8 | 8 | 8 | 8 |
| LUM | >32 | >32 | >32 | >32 | >32 | >32 | >32 |
| TEZ | >32 | >32 | >32 | >32 | >32 | >32 | >32 |
| ELX | 16 | 16 | 16 | 32 | 16 | 16 | 16 |
| ELX-TEZ-IVA | 4/2/3 | 4/2/3 | 4/2/3 | 8/4/6 | 8/4/6 | 8/4/6 | 4/2/3 |

[a]*S. aureus* isolates were grown for 20 h in the presence of serial dilutions of IVA, LUM, TEZ, ELX, or the triple combination of ELX-TEZ-IVA. The concentrations tested ranged from 0.25 $\mu$g/mL to 32 $\mu$g/mL for IVA, LUM, TEZ, and ELX and from 0.5/0.25/0.375 $\mu$g/mL to 64/32/48 $\mu$g/mL for ELX-TEZ-IVA combination. The MIC$_{90}$ was defined as the lowest compound concentration showing a reduction in the optical density at 620 nm of ≈90% in comparison to that for the bacteria grown with the vehicle after 20 h. Each experiment was performed at least two independent times (two technical replicates).

[b]ref, reference strain; early, isolate collected at the early stage of chronic colonization; late, isolate collected after years of persistence, in the advanced stage of chronic colonization.

mouse CFTR (33). However, they can facilitate PK analysis and provide much-needed knowledge on the interactions between CFTR modulators and CF pathogens while avoiding the variables of human studies.

Thus, we aimed to establish whether and how bacterial adaptation affects the antimicrobial activity of CFTR modulators, including ELX and ETI, and the impact of bacterial infection on PK. We designed the study to (i) determine the antimicrobial activity of CFTR modulators and their additive or synergistic effects with antibiotics against *P. aeruginosa* and *S. aureus* reference and clonal isolates collected from CF patients at early and advanced stages of lung colonization; and (ii) evaluate the concentration of IVA, as a model CFTR modulator, in plasma and airway samples and its changes during pneumonia in mice.

## RESULTS

**IVA and ETI showed the most potent antimicrobial activities against longitudinal *S. aureus* isolates, while *P. aeruginosa* isolates were not affected.** To evaluate the impact of bacterial adaptation to the CF lung on the antimicrobial activity of CFTR modulators, we measured minimum inhibitory concentrations (MICs) for clinical clonal isolates collected from CF patients at early and late stages of chronic colonization and previously characterized (see Tables S1 and S2 in the supplemental material) (34–37). We tested single CFTR modulators and the ELX-TEZ-IVA combination, at a ratio of 2:1:1.5, which is used for CF patients (38). IVA showed antimicrobial activity, with MICs ranging from 2 to 8 $\mu$g/mL for all the *S. aureus* isolates (Table 1). The MIC of ELX was 16 $\mu$g/mL for almost all *S. aureus* isolates, except for the J6 isolate (32 $\mu$g/mL). By contrast, the MICs of LUM and TEZ were >32 $\mu$g/mL, indicating little or no effect against any of the *S. aureus* isolates. The ELX-TEZ-IVA combination showed antimicrobial activity against all the *S. aureus* isolates, with MICs ranging from 4/2/3 to 8/4/6 $\mu$g/mL. By contrast, all CFTR modulators showed MICs of >32 $\mu$g/mL against *P. aeruginosa* isolates (Table S3). Our results demonstrated that adaptation does not change either the susceptibility of *S. aureus* strains to IVA and ETI or the resistance of *P. aeruginosa* to CFTR modulators.

**IVA and LUM potentiated the activities of all antibiotics against *S. aureus*.** Next, we determined the potential additive or synergistic effects of the CFTR modulators in combination with antibiotics against reference and clinical *S. aureus* isolates by checkerboard assays. In general, except for TEZ, interaction effects were observed on some isolates regardless of the stage of colonization (Table 2). The fractional inhibitory concentration (FIC) indexes of IVA showed broad-spectrum potentiation of linezolid (LZD) activity (6 out of 7 isolates), while a narrower spectrum was observed for amoxicillin (AMX), vancomycin (VAN), and teicoplanin (TEC). Synergistic effects were also observed between IVA and either VAN or TEC, but only on the Newman reference strain. By contrast, additive effects between either ELX or ETI and the tested antibiotics were more sporadic or completely absent, as in the case of LZD. LUM, which did not show direct antibiotic effects, behaved similarly to IVA in elevating the activities of the tested

**TABLE 2** FIC indexes for IVA, LUM, TEZ, ELX, and the triple combination ELX-TEZ-IVA in combination with common antibiotics against *S. aureus* isolates collected from CF patients[a]

| CFTR modulator | Antibiotic | FIC index for *S. aureus* isolate (stage)[b] | | | | | | |
|---|---|---|---|---|---|---|---|---|
| | | Newman (ref) | A10 (early) | A12 (late) | J6 (early) | J9 (late) | F1 (early) | F5 (late) |
| IVA | AMX | 0.5 | 1 | 2 | 1 | 0.5 | 0.625 | 0.625 |
| | VAN | 0.31 | 0.625 | 0.625 | 1 | 1 | 1 | 1 |
| | TEC | 0.31 | 1 | 1 | 1 | 0.75 | 1 | 0.75 |
| | LZD | 0.75 | 2 | 0.625 | 0.625 | 0.625 | 0.625 | 0.625 |
| | AZM | 0.531 | 2 | 2 | 2 | 2 | 2 | 2 |
| LUM | AMX | 0.625 | 0.625 | 2 | 1 | 0.625 | 0.625 | 0.625 |
| | VAN | 0.155 | 0.625 | 0.75 | 1 | 1 | 1 | 0.625 |
| | TEC | 0.185 | 0.625 | 0.625 | 2 | 2 | 0.625 | 0.75 |
| | LZD | 0.625 | 2 | 0.625 | 2 | 0.75 | 0.625 | 0.625 |
| | AZM | 0.75 | 1 | 1 | 2 | 2 | 1 | 2 |
| TEZ | AMX | 2 | 2 | 2 | 2 | 2 | 2 | 2 |
| | VAN | 2 | 2 | 2 | 2 | 2 | 2 | 2 |
| | TEC | 2 | 2 | 2 | 2 | 2 | 2 | 2 |
| | LZD | 2 | 2 | 2 | 2 | 2 | 2 | 2 |
| | AZM | 2 | 2 | 2 | 2 | 1 | 2 | 2 |
| ELX | AMX | 2 | 0.5 | 0.75 | 2 | 0.75 | 2 | 2 |
| | VAN | 2 | 0.563 | 2 | 2 | 2 | 2 | 2 |
| | TEC | 2 | 2 | 2 | 2 | 2 | 2 | 2 |
| | LZD | 2 | 2 | 2 | 2 | 2 | 2 | 2 |
| | AZM | 0.75 | 1 | 1 | 0.75 | 0.75 | 0.5 | 2 |
| ETI | AMX | 2 | 0.75 | 1 | 2 | 0.75 | 2 | 2 |
| | VAN | 2 | 2 | 2 | 2 | 2 | 2 | 2 |
| | TEC | 2 | 2 | 2 | 0.625 | 2 | 1 | 2 |
| | LZD | 2 | 2 | 2 | 2 | 2 | 2 | 2 |
| | AZM | 2 | 2 | 2 | 2 | 2 | 2 | 2 |

[a]*S. aureus* isolates were grown for 20 h in the presence of serial dilutions of IVA, LUM, TEZ, ELX, or the triple combination ETI, along with the antibiotic AMX, VAN, TEC, LZD, or AZM. The concentrations of the CFTR modulators tested ranged from 0.5 to 32 $\mu$g/mL for IVA, LUM, and ELX and from 0.5/0.25/0.375 $\mu$g/mL to 32/16/24 $\mu$g/mL for the triple combination of ELX-TEZ-IVA (ETI), while antibiotics started from a concentration two-fold greater than the MIC$_{90}$ for the isolate to 32 times below the MIC$_{90}$ of the antibiotic alone. FIC indexes were calculated as (MIC$_{90}$ of CFTR modulator in combination/MIC$_{90}$ of CFTR alone) + (MIC$_{90}$ of antibiotic in combination/MIC$_{90}$ of antibiotic alone). Each experiment was performed at least two independent times (two technical replicates).

[b]A FIC index of <0.5 indicated synergy (dark gray shading), a FIC index of ≥0.50 and <1.0 indicated an additive effect (light gray shading), and a FIC index of ≥1.0 and ≤4 indicated indifference (no shading). ref, reference strain; early, isolate collected at the early stage of chronic colonization; late, isolate collected after years of persistence, in the advanced stage of chronic colonization.

antibiotics and showed better synergistic effects with VAN or TEC against the Newman strain. These findings indicated that the CFTR modulators, mainly IVA and LUM, can enhance the performance of antibiotics used against *S. aureus* infections in CF patients. The potentiating effects appeared to be bacterial isolate dependent, but no correlation with the stage of chronic colonization was found.

**IVA and ETI potentiated the activities of only selected antibiotics against *P. aeruginosa*.** Despite the lack of intrinsic antibacterial activity of the CFTR modulators against *P. aeruginosa* (Table S3), we tested whether they enhanced the activity of antibiotics (Table 3). Strikingly, IVA was found to strongly enhance the activities of colistin (CST) and PMB against 95.45% of the isolates. Synergy, rather than additivity, was detected for most of the isolates. No effect was observed when IVA was combined with ciprofloxacin (CIP), meropenem (MER), or tobramycin (TOB). LUM, TEZ, and ELX showed only a few cases of additive or synergistic effects with selected antibiotics. For ETI, the FIC values indicated an additive effect on the activities of CST and PMB in several isolates, with synergy against the AA2 isolate reached with both antibiotics. In only two isolates did ETI potentiate the activity of CIP. No CFTR modulators affected azithromycin (AZM) activity on *P. aeruginosa* isolates. Notably, no negative interaction of the CFTR modulators with the antibiotics was observed. These findings indicated that IVA and ETI can enhance antibiotic activity, particularly that of CST and PMB, against *P. aeruginosa* in an isolate-dependent manner and independently of the stage of colonization.

**IVA was differentially distributed in plasma, lung, and ELF, with concentrations affected by *P. aeruginosa* infection.** To establish the impact of bacterial infection on CFTR modulator PK, we exploited the mouse model of *P. aeruginosa* pneumonia (39)

**TABLE 3** FIC indexes for IVA, LUM, TEZ, ELX, and ETI in combination with common antibiotics against *P. aeruginosa* isolates collected from CF patients[a]

| CFTR modulator | Antibiotic | FIC index for *P. aeruginosa* isolate (stage)[b] | | | | | | | | | | |
|---|---|---|---|---|---|---|---|---|---|---|---|---|
| | | PAO1 (ref) | AA2 (early) | AA43 (late) | AA44 (late) | MF1 (early) | MF51 (late) | KK1 (early) | KK2 (early) | KK71 (late) | KK72 (late) | RP73 (late) |
| IVA | CIP | 2 | 2 | 2 | 2 | 2 | 2 | 2 | 2 | 2 | 2 | 2 |
| | MER | 2 | 2 | 2 | 2 | 2 | 2 | 2 | 2 | 2 | 2 | 2 |
| | TOB | 2 | 2 | 2 | 2 | 2 | 2 | 2 | 2 | 2 | 2 | 2 |
| | CST | 0.531 | 0.156 | 0.281 | 0.281 | 0.531 | 0.531 | 2 | 0.531 | 0.312 | 0.281 | 0.281 |
| | PMB | 0.562 | 0.281 | 0.281 | 0.281 | 0.281 | 0.281 | 0.375 | 0.516 | 0.312 | 0.281 | 0.281 |
| | AZM | 2 | 2 | 2 | 2 | 2 | 2 | 2 | 2 | 2 | 2 | 2 |
| LUM | CIP | 2 | 0.625 | 2 | 0.5 | 0.281 | 0.75 | 0.625 | 1 | 2 | 2 | 2 |
| | MER | 2 | 2 | 2 | 0.5 | 2 | 2 | 2 | 1 | 2 | 2 | 2 |
| | TOB | 1 | 2 | 2 | 0.5 | 2 | 2 | 1 | 0.75 | 1 | 1 | 0.75 |
| | CST | 1 | 0.625 | 2 | 2 | 2 | 2 | 2 | 2 | 0.5 | 0.375 | 2 |
| | PMB | 2 | 2 | 2 | 2 | 2 | 2 | 2 | 2 | 2 | 2 | 2 |
| | AZM | 2 | 2 | 2 | 2 | 2 | 2 | 2 | 2 | 2 | 2 | 2 |
| TEZ | CIP | 2 | 2 | 2 | 2 | 2 | 2 | 2 | 2 | 2 | 2 | 2 |
| | MER | 2 | 2 | 2 | 2 | 2 | 2 | 2 | 2 | 2 | 2 | 2 |
| | TOB | 2 | 2 | 2 | 2 | 2 | 2 | 2 | 2 | 2 | 2 | 2 |
| | CST | 2 | 2 | 2 | 2 | 2 | 2 | 2 | 0.75 | 2 | 2 | 2 |
| | PMB | 2 | 0.562 | 2 | 2 | 2 | 2 | 2 | 2 | 2 | 2 | 2 |
| | AZM | 2 | 2 | 2 | 2 | 2 | 2 | 2 | 2 | 2 | 2 | 2 |
| ELX | CIP | 2 | 2 | 2 | 2 | 0.687 | 2 | 2 | 2 | 2 | 2 | 2 |
| | MER | 2 | 2 | 2 | 2 | 2 | 1 | 2 | 2 | 2 | 2 | 2 |
| | TOB | 2 | 2 | 2 | 2 | 2 | 2 | 2 | 2 | 2 | 2 | 2 |
| | CST | 2 | 0.562 | 2 | 2 | 2 | 2 | 2 | 2 | 2 | 2 | 2 |
| | PMB | 2 | 2 | 2 | 2 | 2 | 2 | 2 | 2 | 2 | 2 | 2 |
| | AZM | 2 | 2 | 2 | 2 | 2 | 2 | 2 | 2 | 2 | 2 | 2 |
| ETI | CIP | 2 | 2 | 2 | 2 | 0.438 | 1.25 | 2 | 0.594 | 1.25 | 2 | 2 |
| | MER | 2 | 2 | 2 | 2 | 2 | 2 | 2 | 2 | 2 | 2 | 2 |
| | TOB | 2 | 2 | 2 | 2 | 2 | 2 | 2 | 2 | 2 | 2 | 2 |
| | CST | 0.523 | 0.273 | 0.523 | 0.547 | 0.547 | 2 | 2 | 2 | 0.594 | 0.547 | 2 |
| | PMB | 2 | 0.297 | 0.512 | 2 | 2 | 0.547 | 2 | 2 | 0.547 | 0.547 | 2 |
| | AZM | 2 | 2 | 2 | 2 | 2 | 2 | 2 | 2 | 2 | 2 | 2 |

[a]*P. aeruginosa* isolates were grown for 20 h in the presence of serial dilutions of IVA, LUM, TEZ, ELX, or the triple combination of ELX-TEZ-IVA (ETI) along with CIP, MER, CST, TOB, PMB, or AZM. The concentrations of the CFTR modulators tested ranged from 0.5 $\mu$g/mL to 32 $\mu$g/mL for IVA, LUM, TEZ, and ELX and from 0.5/0.25/0.375 $\mu$g/mL to 32/16/24 $\mu$g/mL for the triple combination ETI, while those of the antibiotics started from concentrations 2-fold greater than the $MIC_{90}$ for the isolate to 32 times below the $MIC_{90}$ of the antibiotic alone. FIC indexes are indicated and were calculated as ($MIC_{90}$ of CFTR modulator in combination/$MIC_{90}$ CFTR alone) + ($MIC_{90}$ antibiotic in combination/$MIC_{90}$ antibiotic alone). Each experiment was performed at least two independent times (two technical replicates).

[b]A FIC index of <0.5 indicated synergy (dark gray shading), a FIC index of ≥0.50 and <1.0 indicated an additive effect (light gray shading), and a FIC index of ≥1.0 and ≤4 indicated indifference (no shading). ref, reference strain; early, isolate collected at the early stage of chronic colonization; late, isolate collected after years of persistence, in the advanced stage of chronic colonization.

**TABLE 4** Percentages of IVA and its metabolites M1 and M6 as protein-bound and free active compounds in murine plasma and lungs[a]

| Sample | Protein binding (%) | | |
|---|---|---|---|
| | IVA | M1 | M6 |
| Plasma | | | |
| Bound fraction | 99.97 | 99.89 | 99.47 |
| Free active fraction | 0.03 | 0.11 | 0.53 |
| | | | |
| Lung | | | |
| Bound fraction | 99.74 | 97.17 | 80.3 |
| Free active fraction | 0.26 | 2.83 | 19.7 |

[a]Blood was collected from C57BL/6NCrlBR male mice (8 to 10 weeks of age) and processed to obtain plasma. Lungs were excised, homogenized, and centrifuged, and the supernatants were collected. Protein binding was measured by the dialysis method with rapid equilibrium dialysis inserts, with undiluted plasma or lung homogenate from 2-h samples derived from non-infected untreated mice. Samples were diluted 1:2 in phosphate-buffered saline (PBS) buffer spiked with a solution containing the three analytes against PBS buffer. Samples were analyzed with UPLC-MS/MS in positive multiple-reaction monitoring modes. Values are the means of three replicates.

and evaluated IVA concentrations in murine plasma, lung, and ELF. A mouse model of acute infection established with *P. aeruginosa* reference strain PAO1 was used to mimic early treatment in patients with CF. We selected IVA as the model of a CFTR modulator, since it can be administered as a single compound to patients with CF and it showed the most striking antimicrobial activity *in vitro*. Mice were administered IVA at a dose approximately reflecting that used in humans, as detailed in Material and Methods. In addition, *P. aeruginosa* growth is not influenced by IVA and was therefore selected as a pathogenic model, to avoid potential confounding effects. Indeed, we did not observe any difference in the bacterial loads between the lungs of mice treated with IVA and those treated with the vehicle (Fig. S1). We first determined the protein binding of IVA and its metabolites M1 (pharmacologically active) and M6 (inactive) (40). The percentages of protein binding were very high in the plasma, with those of IVA, M1, and M6 reaching ≥99.5% (Table 4). Similar protein binding was observed in the lung for IVA (99.7%) and M1 (97.2%), while that of M6 was lower (80.3%).

Next, the time-concentration curves of IVA and its metabolites showed that the parent drug accounted for the majority of the total drug, followed by M1 and then M6, in the plasma of both infected and non-infected mice (Fig. 1A). At early time points, IVA and M1 levels were lower in *P. aeruginosa*-infected mice than in non-infected mice (*P* = 0.057).

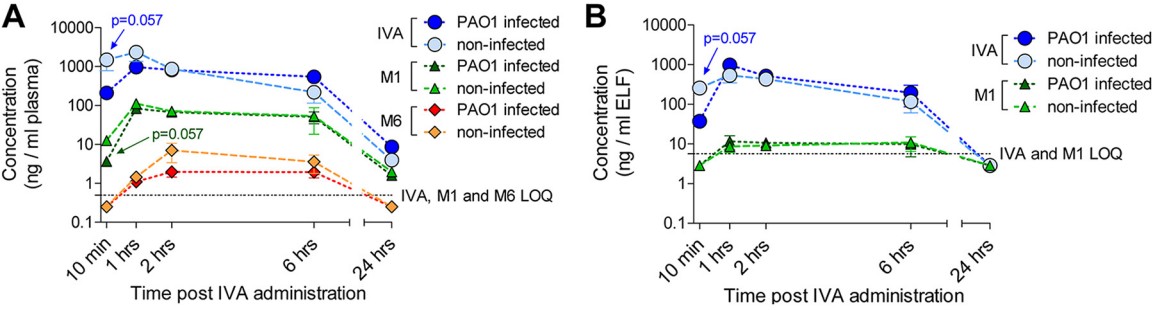

**FIG 1** Ivacaftor, M1, and M6 concentrations in murine plasma and ELF. (A) C57BL/6NCrlBR male mice (8 to 10 weeks of age) were infected with 1 × 10⁶ CFU of *P. aeruginosa* PAO1 by intratracheal administration. A non-infected control group was also tested in parallel. Thirty minutes after infection, the mice were treated with 3 mg/kg IVA in 10% PEG 400, 10% Tween 80, and 80% saline by intraperitoneal administration. Mice were sacrificed at 10 min and 1, 2, 6, and 24 h after IVA administration. Blood was collected and processed to obtain plasma. IVA, M1, and M6 concentrations in plasma were measured by HPLC-MS/MS. (B) BALF was collected and centrifuged, and the supernatant was used to quantify the IVA concentration. The volume of the ELF was determined by using the ratio of the urea concentration in BALF to that in plasma. The amounts of IVA and M1 were measured by HPLC-MS/MS. Data (derived from 3 to 4 mice) are represented as the mean values ± standard errors of the means (SEMs). Limits of quantification (LOQ) for IVA and its metabolites are indicated. A value corresponding to LOQ/2 was assigned to undetectable samples at specific time points (e.g., 10 min and/or 24 h). Statistical significance was calculated by the Mann-Whitney test comparing the infected and non-infected mouse groups at each time point.

**TABLE 5** Pharmacokinetic parameter estimates for IVA in plasma, ELF, and lungs from infected and non-infected mice[a]

| Sample | Mouse group | Pharmacokinetic parameters[b] | | | | | | | |
|---|---|---|---|---|---|---|---|---|---|
| | | $t_{max}$ (h) | $C_{max}$ (ng/mL) | $t_{last}$ (h) | $C_{last}$ (h) | $AUC_{last}$ (h·ng/mL) | $AUC_{inf}$ (h·ng/mL) | $t_{1/2}$ (h) | MRT (h) |
| Plasma | Non-infected | 1 | 2,345 | 24 | 4.0 | 7,482 | 7,498 | 2.9 | 2.9 |
| | Infected | 1 | 968 | 24 | 8.6 | 9,111 | 9,152 | 3.3 | 4.7 |
| ELF | Non-infected | 1 | 537 | 6 | 116.7 | 1,932 | 2,307 | 2.2 | 2.1 |
| | Infected | 1 | 981 | 6 | 194.3 | 2,590 | 3,236 | 2.3 | 2.2 |
| Lung | Non-infected | 1 | 249 | 6 | 68.2 | 946 | nd | nd | 2.2 |
| | Infected | 1 | 364 | 24 | 4.4 | 4,010 | 4,033 | 3.7 | 5.2 |

[a]C57BL/6NCrlBR male mice (8 to 10 weeks of age) were infected with $1 \times 10^6$ CFU of *P. aeruginosa* PAO1 by intratracheal administration. A non-infected control group was also tested in parallel. Thirty minutes after infection, the mice were treated with 3 mg/kg IVA in 10% PEG 400, 10% Tween 80, and 80% saline by intraperitoneal administration. Mice were sacrificed at 10 min and 1, 2, 6, and 24 h after IVA administration. Blood was collected and processed to obtain plasma. BALF was collected and centrifuged, and the supernatant was used to quantify the IVA concentration. Lungs were excised, homogenized, and centrifuged, and the supernatants were used to quantify IVA concentrations. Plasma, lung homogenate, and BALF were added to a Phree phospholipid removal plate (Phenomenex) with acetonitrile and 0.1% formic acid in order to eliminate phospholipids, decreasing the matrix effect. Eluates were analyzed by UPLC-MS/MS with a linear gradient in multiple reaction monitoring positive mode. Calibration ranges were 0.5 to 750 ng/mL in plasma and BALF and 4 to 1,000 ng/g in the lung homogenate. Concentrations of IVA in ELF were determined using the ratio of the urea concentration in plasma to that in BALF: concentration in ELF = (drug concentration in BALF) × (urea in plasma)/(urea in BALF). The data are the geometric means of values from 3 to 4 mice.
[b]$t_{max}$, time of maximum concentration; $C_{max}$, maximum concentration; $t_{last}$, time of last quantifiable concentration; $C_{last}$, last quantifiable concentration; $AUC_{last}$, AUC to the last quantifiable concentration level; $AUC_{inf}$, AUC to infinity; $t_{1/2}$, half-life; MRT, mean residence time; nd, not determined.

However, at later time points, the IVA levels were slightly higher in infected mice than in non-infected mice. M6 levels ranged from low to undetectable in the plasma of all mice. Peak plasma levels of IVA, which were reached 1 h after treatment, were higher in non-infected compared to infected mice (Table 5), as were M1 levels (Table S4). Nonetheless, the area under the curve (AUC) for the period from 10 min ($t_{10}$) to the last quantifiable concentration ($AUC_{last}$) and the AUC to infinity ($AUC_{inf}$) for IVA were higher in *P. aeruginosa*-infected than non-infected mice, indicating higher exposure in the presence of infection. This was confirmed by the higher last quantifiable concentration ($C_{last}$) at 24 h, half-life ($t_{1/2}$), and mean residence time (MRT) in infected compared to non-infected mice. By contrast, M1 AUC, $C_{last}$, $t_{1/2}$, and MRT values were similar in both groups.

The IVA profile in the ELF followed that in the plasma samples up to 6 h, with higher levels in non-infected mice than in infected mice soon after treatment ($P = 0.057$) (Fig. 1B). *P. aeruginosa* infection resulted in higher IVA peak levels and $AUC_{last}$, $AUC_{inf}$, and $C_{last}$ levels compared to levels in non-infected mice (Table 5). The partitioning to the ELF was estimated by comparing the AUC of the unbound compound (fAUC) in the plasma to that in the ELF. For the calculation of fAUC, a factor of 0.03% was used for the IVA parent drug in the plasma (Table 4). The ELF/plasma fAUC ratio of IVA was high, indicating that IVA penetrates ELF well (Table 6). M1 was only detected between 1 and 6 h post-administration in ELF of infected and non-infected mice (Fig. 1B). The ELF/plasma fAUC ratio was also high for M1, indicating high penetration in the ELF. M6 was not quantifiable for any conditions.

The IVA profile in the lung homogenates showed similar levels in infected and non-infected mice up to 2 h, while the levels were higher in *P. aeruginosa*-infected mice after 6 h (Fig. S2). The lung/plasma fAUC ratio of IVA indicated that IVA penetrates the lung very well, particularly in infected mice (Table 6). M1 showed a profile in the lung similar to that

**TABLE 6** Penetration of IVA in ELF and lungs[a]

| | PK value by sample type | | | | | | PK ratios[b] | | | |
|---|---|---|---|---|---|---|---|---|---|---|
| | Plasma | | Lung | | ELF | | Lung/plasma | | ELF/plasma | |
| Mouse group | $fC_{max}$ (ng/mL) | fAUC (h·ng/mL) | $fC_{max}$ (ng/mL) | fAUC (h·ng/mL) | Cmax (ng/mL) | AUC (h·ng/mL) | $C_{max}$ | AUC | $C_{max}$ | AUC |
| Non-infected | 0.70 | 2.20 | 0.65 | 2.44 | 537 | 1,910 | 0.93 | 1.11 | 767 | 868 |
| Infected | 0.29 | 2.72 | 0.94 | 10.62 | 981 | 2,587 | 3.24 | 3.90 | 3,383 | 951 |

[a]The $C_{max}$ and AUC of plasma and lung homogenates were corrected for the free fraction according to the results of the protein binding experiment (0.03% for plasma and 0.26% for lung; see Table 4) to obtain the $C_{max}$ and AUC of unbound IVA ($fC_{max}$ and fAUC). Data are the geometric means of values from 3 to 4 mice.
[b]Lung/plasma values are the ratio of lung free maximum concentration or lung exposure to free maximum concentration divided by those in plasma, and ELF/plasma values are the ratio of ELF maximum concentration or ELF exposure to maximum concentration divided by plasma free maximum concentration or plasma exposure to free maximum concentration. $fC_{max}$ and fAUC refer to unbound IVA in the plasma and lung.

in the ELF in both infected and non-infected mice (Fig. S2 and Tables S4 and S5). M6 was not quantifiable in either infected or non-infected mice at any time point.

These findings indicated that the IVA parent drug, accounting for the majority of the total drug, readily distributes to the airways and is higher during *P. aeruginosa* infection.

## DISCUSSION

CFTR modulators correct the molecular defect underlying CF and disease manifestations. Since bacterial lung infections or colonization are one of the hallmarks of CF disease, the effect (if any) of CFTR modulators on bacteria could deeply affect the course of the disease. Here, we tested the impact of bacterial adaptation on the antimicrobial activity of CFTR modulators and that of bacterial infection on PK. We found that (i) IVA and the newly licensed ETI had the most potent intrinsic antimicrobial activity against longitudinal *S. aureus* isolates, while *P. aeruginosa* isolates were not affected; (ii) IVA and LUM potentiated the activities of all antibiotics against *S. aureus*, while IVA and ETI potentiated in particular that of polymyxins against *P. aeruginosa*; (iii) antimicrobial activity of CFTR modulators was independent of the stage of colonization for both bacterial species; and (iv) *P. aeruginosa* infection affected the distribution of IVA to plasma, lung, and ELF.

Our study, focused on previously characterized clinical clonal isolates collected from CF patients at early and late stages of chronic colonization, showed that IVA has a spectrum of relevant antibacterial activities that spans both early and late isolates of *S. aureus*. In addition, ETI is active against *S. aureus*, and its activity seems to be mainly driven by IVA, since the effective concentration of IVA in the triple combination was in the range of that as a single drug. In addition, we showed for the first time that ELX has broad antibacterial activity against *S. aureus*. Notably, the anti-*S. aureus* activity of CFTR modulators was independent of the stage of colonization. These activities deserve further investigation with additional biobanks, including those from CF patients under CFTR modulator treatment. Regarding CFTR modulator and antibiotic interactions, we expanded the notion that IVA can increase the activity of several classes of antibiotics against *S. aureus*. Furthermore, we showed that LUM, with no intrinsic antibacterial activity against *S. aureus*, can positively interact with several classes of antibiotics. The specific mechanism underlying the enhancement of antibiotic activities by IVA and LUM against *S. aureus* is unknown and should be investigated.

We observed a different scenario for *P. aeruginosa* clinical clonal isolates collected from CF patients at early and late stages of chronic colonization and previously characterized. IVA showed no antibacterial activity against any *P. aeruginosa* early or late isolates, suggesting that adaptation to the CF environment does not induce modification of bacterial structures or functions potentially causing resistance to IVA. This is in line with other reports showing that IVA is inactive against *P. aeruginosa* and other Gram-negative species (e.g., *Klebsiella pneumoniae* and *Acinetobacter baumannii*) (22). Gram-negative bacteria have a formidable outer membrane barrier, particularly against hydrophobic drugs such as CFTR modulators, and this can explain the absence of IVA activity. It was initially hypothesized that IVA inhibits DNA gyrase and topoisomerase IV, since it has a structural resemblance with quinolones. However, it has recently been demonstrated that its antibacterial activity is not due to a quinolone-like mode of action; rather, it inhibits *P. aeruginosa* cell envelope biogenesis, although this activity was shown only when combined with PMB (24). We also observed that IVA can specifically act in concert with polymyxins, such as CST and PMB, to enhance their antibacterial activities in almost all the isolates tested, while no additive or synergistic effects were observed for the other classes of antibiotics. Similar to IVA, also ETI potentiated the activities of CST and PMB in several isolates. Although the activity of ETI can be driven by IVA, some exceptions have been observed, suggesting potential antagonisms when IVA is combined in the triple combination in an isolate-dependent manner. To our knowledge, while synergies between IVA and antibiotics against *S. aureus* and *P. aeruginosa* have been previously described, no data have been provided for next-generation CFTR modulators, such as ELX and ETI. In addition, previous studies on

IVA antibacterial activity were mainly performed on uncharacterized bacterial isolates. Our study aimed to clarify whether the stage of chronic colonization could affect the antibacterial activities of CFTR modulators. The finding that adaptation to the CF lung did not change the activity of these small molecules indicates their potential impact on *P. aeruginosa* and *S. aureus* during the entire course of colonization.

To further interpret these results, it is critical to know the amounts of CFTR modulators that reach the airway lumen and whether bacterial pneumonia can alter their concentrations. To investigate the PK profile and provide insight into the interactions between CFTR modulators and CF pathogens in the absence of confounding variables of human studies, we exploited a mouse model of *P. aeruginosa* pneumonia. We focused on the early stage of the disease, since IVA is now administered early in life and it is likely that also ETI will be approved for very early treatment. Our animal experiments were performed with PAO1 reference strain, given that no significant differences have been observed in the phenotypic characteristics and *in vivo* responses between reference and early *P. aeruginosa* clinical strains in previous testing (41). *P. aeruginosa* growth was not influenced by IVA and was therefore selected as a pathogenic model, avoiding potential confounding effects. Our results showed that the parent drug accounted for the majority of the total drug in plasma, lung, and ELF, similar to previous reports focused solely on murine plasma, but different from humans, who show higher concentrations of M1 in plasma (26). Interestingly, exposure of both plasma and airways to IVA was higher in mice with acute *P. aeruginosa* lung infection than in non-infected mice. In our study, the IVA $C_{max}$ in the plasma was in the range of a few micrograms per milliliter. However, the protein binding of IVA and its metabolites was extremely high in both the plasma and lung, indicating that the amount of free active compound was low, in agreement with previous studies (26). For instance, the $C_{max}$ of free active IVA in plasma was in the range of approximately 1 ng/mL in our model, similar to the peak free plasma concentration measured in CF patients treated with IVA (27).

When we focused on the airways, IVA showed good penetration into the ELF, reaching higher concentrations than in the plasma. The IVA $C_{max}$ was approximately 0.5 to 1 $\mu$g/mL in the ELF. Notably, we observed antimicrobial activity or additive or synergistic effects with CST at IVA concentrations ranging from 1 to 8 $\mu$g/mL, which were comparable to or moderately higher than those observed in the murine ELF. Schneider and colleagues measured an IVA concentration of 0.15 $\pm$ 0.05 $\mu$g/mL in the sputum of a CF patient at 2.5 h post-IVA administration (29), which would have excluded antimicrobial activities of CFTR modulators in CF patients. However, data from sputa of single patients cannot be considered conclusive, and IVA quantification in a cohort of patients is needed. In addition, IVA protein binding in airway secretions and how mucus impacts IVA availability should be evaluated, since they could extensively affect the activity. On the other hand, IVA is known to accumulate with repeated doses, particularly when it is taken with a fatty meal (42). Our work analyzed IVA concentrations in mice after administering a single dose, but higher levels would be expected during chronic treatment. In this regard, IVA levels following prolonged treatment in CF patients are unknown. Therefore, the antimicrobial activity of IVA in the airways of CF patients could be plausible and deserve further investigation. In addition, our findings support performing new studies on the potential benefits of pulmonary administration of CST in combination with IVA in treating *P. aeruginosa* lung infections (43).

**Conclusions.** This work supports the interaction of CFTR modulators with opportunistic bacterial infection and antibiotics. Our results underline that CFTR modulators influence antibiotic efficacy through a direct or synergistic effect. This may orient the selection of specific antibiotics to treat infection in combination with CFTR modulators. Importantly, CFTR modulator efficacy is not influenced by specific bacterial phenotypes (early versus late adapted), indicating drug targeting at any stage of colonization. Since the bacterial strains used in this study were collected several years ago, they did not show some of the antibiotic resistances that have emerged in CF clinics in recent

years. The challenge ahead is to validate our results with additional biobanks that include longitudinal strains isolated from CF patients under CFTR modulator therapies and with different patterns of antibiotic resistance.

So far, neither animal nor human studies have clarified the impact of infection on concentrations and biodistribution of CFTR modulators. Our results in a mouse model of *P. aeruginosa* pneumonia underline the importance of testing PK during infection, particularly in the airways. The challenge ahead is to improve these studies in mouse models by comparing initial and adapted bacterial isolates and including additional variables, such as polymicrobial infection and the concomitant use of other therapies, to further reflect the complexity of human diseases. Furthermore, these analyses should be extended to chronic infection mouse models with repeated administrations to determine the impacts both of drug accumulation and lung damage on biodistribution. This would help to optimize the drug regimens and improving their efficacy in the view of personalized treatments.

## MATERIALS AND METHODS

**Ethics statement.** The animal studies adhered to the Italian Ministry of Health guidelines for the use and care of experimental animals (IACUC 954). The experiments with CF *P. aeruginosa* and *S. aureus* isolates and storage of biological materials were approved by the Ethics Commissions of Hannover Medical School and University Hospital Münster (Germany).

**Bacterial strains.** *P. aeruginosa* and *S. aureus* strains included reference PAO1 and Newman. CF clinical isolates recovered at early and late chronic colonization were RP73, AA2, AA43, AA44, MF1, MF51, KK1, KK2, KK71, and KK72 for *P. aeruginosa* and A10, A12, J6, J9, F1, and F5 (34–37) for *S. aureus*. Clinical strains were collected at Hannover Medical School and University Hospital Münster (Germany). Phenotype and antimicrobial resistance were previously characterized (see Tables S1 and S2 in the supplemental material) (34–37).

**MIC measurements.** The MICs of CFTR modulators were determined using the broth microdilution susceptibility testing method as described in the supplemental material and elsewhere (34, 44).

**Checkerboard assay.** The synergistic activities of the CFTR modulators combined with TOB, CIP, CST, PMB, MER, and AZM for *P. aeruginosa* and AMX, TEC, LZD, VAN, and AZM for *S. aureus* were determined by the checkerboard method in cation-adjusted Mueller-Hinton broth (45), as detailed in the supplemental material.

**Mouse model.** C57BL/6NCrlBR male mice (8 to 10 weeks of age) were challenged with $1 \times 10^6$ colony forming units (CFU) of planktonic strain PAO1 by intratracheal injection. Mice were treated with 3 mg/kg IVA or vehicle (10% polyethylene glycol 400 [PEG 400], 10% Tween 80, and 80% saline) (20) by intraperitoneal injection half an hour after infection, according to the ARRIVE guidelines (46). IVA dose was calculated based on a single adult dose (150 mg) adjusted for mouse weight, assuming that an adult with CF weighs 50 kg (20). Blood, bronchoalveolar lavage fluid (BALF), and lung samples were recovered at different time points, as described elsewhere (39) for PK and as reported in the supplemental material.

**Protein binding and concentration.** Protein binding was measured in mouse plasma and lung homogenate by rapid equilibrium dialysis. Samples were analyzed by ultraperformance liquid chromatography-tandem mass spectrometry (UPLC-MS/MS). Quantitation was performed via multiple reaction monitoring using the transitions reported in Table S6. Concentrations of IVA and its metabolites in ELF were determined by using the ratio of the urea concentration in BALF to that in plasma (47). Protein binding was expected to be negligible in ELF (47). The drug concentration in ELF was assumed to be equal to the unbound concentration. Additional details are provided in the supplemental material.

**PK evaluation.** PK parameters were estimated using PK Solver Excel with a noncompartmental approach consistent with the intraperitoneal route of IVA administration. The AUC was calculated using the linear trapezoidal method for the period from $t = 0$ to the time of the last quantifiable concentration level ($AUC_{0–last}$). Evaluation of the terminal elimination phase was not practical, as terminal phase concentration data were either not available (low doses) or sparse.

**Statistics.** Statistical analyses were performed with Prism (GraphPad Software, Inc., San Diego, CA, USA) using a nonparametric two-tailed Mann-Whitney $U$ test for compound concentrations and two-way analysis of variance with Bonferroni's multiple-comparison test for CFU. A $P$ value of $\leq 0.05$ was considered statistically significant.

## SUPPLEMENTAL MATERIAL

Supplemental material is available online only.

**SUPPLEMENTAL FILE 1**, PDF file, 0.2 MB.

## ACKNOWLEDGMENTS

We thank Marzia Giustra for her invaluable contribution to the *in vitro* assays and Burkhard Tümmler (Medizinische Hochschule Hannover, Germany) and Barbara C. Kahl (University Hospital Münster, Germany) for supplying, respectively, *P. aeruginosa* and *S.*

*aureus* clinical isolates collected from CF patients. This study was supported by the Italian Cystic Fibrosis Research Foundation (FFC 15/2018 to C.C.), with contribution of the Delegazione FFC di Milano and Gruppo di Sostegno FFC di Morbegno, and to A.B. by the Cystic Fibrosis Foundation (BRAGON19G0). G.M. and F.C. have been Italian CF Research Foundation fellows (FFC 15/2018). The funders had no role in study design, data collection and analysis, decision to publish, or preparation of the manuscript.

Conceiving and Designing the Experiments, C.C. and A.B.; Performing Experiments, C.C., R.G., A.C., B.A.-F., G.M., F.C., C.B., and U.B.; Analyzing Data, C.C. and C.B.; Performing Statistical Analysis, C.C.; Interpretation of Experiment Results, C.C., R.G., A.C., B.A.-F., C.B., G.B., and A.B.; Preparing Figures, C.C.; Writing the Manuscript, C.C. and A.B.; Revising the Manuscript, C.C., G.B. and A.B.

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
