## [Reviewer comments · Microbiology Spectrum]

Microbiology Spectrum

Mutual effects of single and combined CFTR modulators and bacterial infection in cystic fibrosis

Cigana Cristina, Ruggero Gianella, Alice Colavolpe, Beatriz Alcalá-Franco, Giulia Mancini, Francesca Colombi, Chiara Bisogno, Ulla Bastrup, Giovanni Bertoni, and Alessandra Bragonzi

Corresponding Author(s): Alessandra Bragonzi, IRCCS San Raffaele Scientific Institute

Review Timeline:

Submission Date:	October 27, 2022
Editorial Decision:	November 21, 2022
Revision Received:	December 1, 2022
Accepted:	December 9, 2022

Editor: Joanna Goldberg

Reviewer(s): The reviewers have opted to remain anonymous.

Transaction Report:

DOI: <https://doi.org/10.1128/spectrum.04083-22>

November 21, 2022

Dr. Alessandra Bragonzi

Infections and Cystic Fibrosis Unit, Division of Immunology, Transplantation and Infectious Diseases, IRCCS San Raffaele Scientific Institute

Infections and Cystic Fibrosis Unit, Division of Immunology, Transplantation and Infectious Diseases

via Olgettina 58

Milano 20132

Italy

Re: Spectrum04083-22 (Mutual effects of single and combined CFTR modulators and bacterial infection in cystic fibrosis)

Hi Alessandra:

I hope you are doing well and missed seeing you at NACFC.

Thank you for submitting your manuscript to Microbiology Spectrum.

I have reviewed your submission for Microbiology Spectrum and have only a few suggestions that I would like you to try to incorporate into this manuscript. I don't think these will take more than some careful rewording or rewriting. In particular:

Table 1-I could not figure out how the Staphylococcus aureus isolates in this table related to the patients referred to in Table S2. This needs to be made clearer.

Line 146-I would refer back to Table S3 again here.

Line 159 and beyond-I like the idea of testing IVA in the mouse model of pneumonia. However, I wanted a little more rationale in the Results section of the choice of the acute pneumonia model (rather than a chronic model), the selection of Pseudomonas aeruginosa strain PAO1 (you do mention this eventually in the Discussion) and in the amount of IVA given (which you do refer to the Methods). Realize I have no issue with any of this, but I think better justification up front will head off any criticism.

Finally and perhaps most critically, for the IVA distribution, how do you know whether any of the differences you did see with PAO1 in the mice (Figure 1 A and Tables 4-6) are biologically important? Is there any information about how much IVA is needed to see an increase on gating in mice? Or in any system? In other words, there may be an effect due to infection, but does it even make a difference? I am certainly not proposing that you should do this experiment with CFTR knock out mice, but has anyone else added in IVA to these mice to see what doses are needed to see correct gating? Without this type of information, I think you will need to tone down your conclusions and suggest that these findings may only be relevant under the conditions that you tested (not other strains, doses of PAO1, doses of IVA, etc.).

Minor suggestion-In the headers for the various Results sections, the first time you use abbreviations, please write out the whole words and then put the abbreviations in parentheses. You had this in the tables, but nowhere else. Even I had to look back to remind myself of what an ELF was (not a hobbit :).

Sincerely,

Joanna

Link Not Available

ASM policy requires that data be available to the public upon online posting of the article, so please verify all links to sequence

records, if present, and make sure that each number retrieves the full record of the data. If a new accession number is not linked or a link is broken, provide production staff with the correct URL for the record. If the accession numbers for new data are not publicly accessible before the expected online posting of the article, publication of your article may be delayed; please contact the ASM production staff immediately with the expected release date.

Sincerely,

Joanna Goldberg

Journals Department
Reviewer comments:

Staff Comments:

Preparing Revision Guidelines

Please return the manuscript within 60 days; if you cannot complete the modification within this time period, please contact me. If you do not wish to modify the manuscript and prefer to submit it to another journal, please notify me of your decision immediately so that the manuscript may be formally withdrawn from consideration by Microbiology Spectrum.

Response to editor:

Q1. Table 1-I could not figure out how the *Staphylococcus aureus* isolates in this table related to the patients referred to in Table S2. This needs to be made clearer.

R1. *Thanks for the suggestion. We have modified Table S2 including the isolate name.*

Q2. Line 146-I would refer back to Table S3 again here.

R2. *We introduced the reference to Supplementary Table S3.*

Q3. Line 159 and beyond-I like the idea of testing IVA in the mouse model of pneumonia. However, I wanted a little more rationale in the Results section of the choice of the acute pneumonia model (rather than a chronic model), the selection of *Pseudomonas aeruginosa* strain PAO1 (you do mention this eventually in the Discussion) and in the amount of IVA given (which you do refer to the Methods). Realize I have no issue with any of this, but I think better justification up front will head off any criticism.

R3. *We thank the editor and agree with this suggestion. In agreement, we have amended the manuscript including details in different sections.*

*Line 162 page 6: "A mouse model of acute infection established with *P. aeruginosa* reference strain PAO1 was used to mimic early treatment in patients with CF."*

Line 165 page 6: "Mice were administered IVA at a dose approximately reflecting that used in humans as detailed in Material and Methods."

In addition, we comment on the usefulness of extending the study to the chronic infection model in the conclusions.

Line 310 page 11. "Furthermore, these analyses should be extended to chronic infection mouse models with repeated administrations to determine the impact both of drug accumulation and lung damage on biodistribution."

Q4. Finally and perhaps most critically, for the IVA distribution, how do you know whether any of the differences you did see with PAO1 in the mice (Figure 1 A and Tables 4-6) are biologically important? Is there any information about how much IVA is needed to see an increase on gating in mice? Or in any system? In other words, there may be an effect due to infection, but does it even make a difference? I am certainly not proposing

OSPEDALE SAN RAFFAELE
ISTITUTO DI RICOVERO E CURA A CARATTERE SCIENTIFICO

that you should do this experiment with CFTR knock out mice, but has anyone else added in IVA to these mice to see what doses are needed to see correct gating? Without this type of information, I think you will need to tone down your conclusions and suggest that these findings may only be relevant under the conditions that you tested (not other strains, doses of PAO1, doses of IVA, etc.).

R4. *Thank you for your comment. We clarify in the introduction the concept that mouse and human CFTR are different. Thus, mouse models limit efficacy evaluation on CFTR but they are still instrumental for PK studies as investigated in our paper. For these reasons, we do not believe that CFTR mutant mice are relevant.*

Line 99 page 3. "Mouse models are not helpful for efficacy studies, since cross-species comparative studies have highlighted differences between human and mouse CFTR (33). However, they can facilitate PK analysis and provide much-needed knowledge on the interactions between CFTR modulators and CF pathogens avoiding variables of human studies."

Regarding the biological importance of IVA biodistribution in relation to infection, we reported on line 277 page 10: "Notably, we observed antimicrobial activity or additive/synergistic effects with CST at IVA concentrations ranging from 1 to 8 $\mu\text{g/ml}$, which were comparable to or moderately higher than those observed in the murine ELF."

These results suggest the biological relevance of our findings. However, we point out the challenge ahead to further translate our data to humans in the conclusions sections.

Q5. Minor suggestion-In the headers for the various Results sections, the first time you use abbreviations, please write out the whole words and then put the abbreviations in parentheses. You had this in the tables, but nowhere else. Even I had to look back to remind myself of what an ELF was (not a hobbit :).

R5. *We included these corrections.*

December 9, 2022

Dr. Alessandra Bragonzi
IRCCS San Raffaele Scientific Institute
Infections and Cystic Fibrosis Unit, Division of Immunology, Transplantation and Infectious Diseases
via Olgettina 58
Milano 20132
Italy

Re: Spectrum04083-22R1 (Mutual effects of single and combined CFTR modulators and bacterial infection in cystic fibrosis)

Hi Alessandrai:

I am happy to give you an early but not unexpected holiday present...your paper has been accepted to Microbiology Spectrum.

I wish you and your friends, family, and colleagues a warm holiday season and a happy healthy productive new year.

Sincerely,
Joanna

I will forward your paper on to the ASM Journals Department for publication. You will be notified when your proofs are ready to be viewed.

Sincerely,

Joanna Goldberg
Editor, Microbiology Spectrum
